# Scaling growth rates for perovskite oxide virtual substrates on silicon

Jason Lapano[1], Matthew Brahlek[1,5], Lei Zhang [2], Joseph Roth[1], Alexej Pogrebnyakov[1] & Roman Engel-Herbert[1,3,4]

The availability of native substrates is a cornerstone in the development of microelectronic technologies relying on epitaxial films. If native substrates are not available, virtual substrates - crystalline buffer layers epitaxially grown on a structurally dissimilar substrate - offer a solution. Realizing commercially viable virtual substrates requires the growth of high-quality films at high growth rates for large-scale production. We report the stoichiometric growth of $SrTiO_3$ exceeding $600 \, nm \, hr^{-1}$. This tenfold increase in growth rate compared to $SrTiO_3$ grown on silicon by conventional methods is enabled by a self-regulated growth window accessible in hybrid molecular beam epitaxy. Overcoming the materials integration challenge for complex oxides on silicon using virtual substrates opens a path to develop new electronic devices in the More than Moore era and silicon integrated quantum computation hardware.

[1] Department of Materials Science and Engineering, Pennsylvania State University, University Park, PA 16802, USA. [2] Department of Materials Science and Engineering, University of California, Berkeley, CA 94720, USA. [3] Department of Physics, Pennsylvania State University, University Park, PA 16802, USA. [4] Department of Chemistry, Pennsylvania State University, University Park, PA 16802, USA. [5] Present address: Materials Science and Technology, Oak Ridge National Lab, Oak Ridge, TN 37831, USA. Correspondence and requests for materials should be addressed to R.E.-H. (email: rue2@psu.edu)

Successful implementation of epitaxial thin film technologies hinges upon the availability of an economical single-crystal substrate. These substrates must be both chemically and structurally compatible with the desired film to prevent unwanted interfacial defect formation, and must be economically feasible for large-scale production. When no native single crystalline substrate that satisfies these requirements is available, virtual substrates, i.e., the growth of buffer layers allowing a change in lattice parameter, structure or even chemistry of the available substrate, can provide a solution. In such cases robust and cost effective, while scalable, material integration schemes that suffice stringent economic requirements are in demand to realize new device generations with improved performance at lower cost, weight, and size. Such metamorphic epitaxial materials also represent a route to expand the application space of existing devices, and to realize completely new technologies by stabilizing material phases with otherwise unattainable properties. The technical challenges, such as scalability, high-throughput[1] and compatibility of the individual fabrication steps to exploit such advancements and deploy them into the market impose stringent conditions for the entire materials integration process.

Metamorphic buffer layers have proven their usefulness at an industrial scale for SiGe, III–V compound semiconductors, and group III-nitrides, setting record performance in a wide application space from photovoltaics using a metamorphic multijunction solar cell with efficiencies up to 46%[2–4], III–V based lasers monolithically integrated on Si[5,6], high-electron mobility transistors[7–9], and heterojunction tunnel field effect transistors for high-performance low-power logic[10]. Metamorphic epitaxial materials have allowed for more cost effective and larger non-native substrates for electronic devices[11,12], and are being considered key to realize and advance the existing quantum computing materials platform of Si/SiGe heterostructures towards all-electrical control of Si-based qubits[13].

A metamorphic buffer technology for complex perovskite oxides on Si is highly desirable given their wide range of properties, which can even be expanded utilizing epitaxial strain[14]. Although epitaxial integration of $SrTiO_3$ on Si has been successfully demonstrated[15] and has been scaled up to 200 mm Si wafers using an industry-scale molecular beam epitaxy (MBE) system[16], typical film growth rates of about 50 nm h$^{-1}$ or lower[16–19] impede high-throughput required for profitability. MBE systems have been operated on an industrial scale since the 1980s[20] primarily for the growth of high-quality binary semiconductor structures with controlled stoichiometry and desired composition at growth rates in excess of 1000 nm h$^{-1}$[21,22]. Metamorphic buffers have been developed for nonoxide materials using this growth technology[23,24]. The key feature permitting such high-quality material by MBE is the adsorption-controlled growth kinetics. The more volatile element is supplied in excess and desorbs in a self-regulated fashion if not incorporated into the film, which dramatically simplifies the flux calibration and enhances reproducibility of the growth process[25,26]. Therefore, the roadblock to integrate perovskites oxide materials by MBE or any other thin film growth technique in a scalable, economic way is the lack of an adsorption-controlled growth mechanism at high film growth rate. Specifically in the case of $SrTiO_3$ deviations from ideal stoichiometry result in a lattice parameter expansion to accommodate defects formed in the film when grown under nonstoichiometric conditions[17,27,28].

Commonly employed thin film growth techniques can in principle be scaled up to achieve wafer size conformity for complex perovskite oxides and has been demonstrated for pulsed laser deposition (PLD)[29], sputtering, metalorganic chemical vapor deposition[30], atomic layer deposition, and MBE[16]. However, scaling up growth rates have resulted in nonstoichiometric

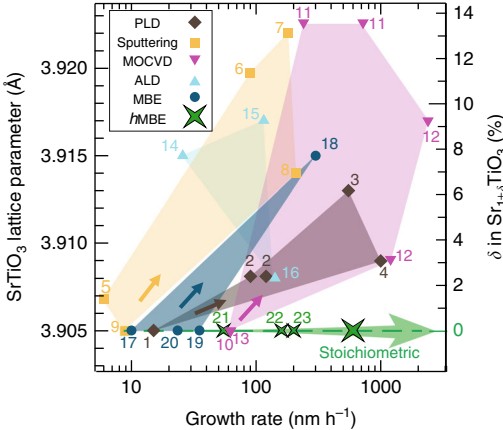

**Fig. 1** Growth rates and control of film stoichiometry. Intrinsic $SrTiO_3$ film lattice parameter reported for different growth rates using scalable oxide thin film growth techniques: pulsed laser deposition (PLD), sputtering, metal-organic chemical vapor deposition (MOCVD), atomic layer deposition (ALD), molecular beam epitaxy (MBE), and hybrid molecular beam epitaxy (hMBE). In all cases except hybrid MBE the degree of film nonstoichiometry increased with growth rate. The number close to data points shown refer to references detailed in Supplementary Fig. 1. The defect concentration $\delta$ of Sr-rich ($Sr_{1+\delta}TiO_3$) and Ti-rich ($Sr_{1-\delta}TiO_3$) films due to nonstoichiometric growth condition was determined from the intrinsic film lattice parameter expansion using a calibration curve given in Supplementary Fig. 2. A growth rate of 0.6 µm h$^{-1}$ for stoichiometric $SrTiO_3$ by hybrid MBE was achieved for the growth on LSAT substrates

$SrTiO_3$ films. Figure 1 compiles the degree of stoichiometry control, quantified by the intrinsic $SrTiO_3$ film lattice parameter expansion, as a function of growth rate for different thin film growth methods. Further details can be found in Supplementary Figs. 1 and 2. Sputtering and PLD utilize targets where the indirect control of deposition parameters makes stoichiometry control more challenging with increasing film growth rate. Conventional solid-source MBE growth for perovskite oxides requires precise calibration of the Sr and Ti fluxes, however, the level of oxygen pressure needed for high-growth rates adversely affect effusion cell flux stability due to unintentional oxidation of the source material in the crucible[31], thus limiting this technique to low growth rates[16,17] and precluding it as a cost-effective production tool for virtual perovskite oxide substrates[17,32].

We report that the adsorption-controlled growth of $SrTiO_3$ films by hybrid MBE is scaled to growth rates in excess of 600 nm h$^{-1}$ with no degradation in structural film quality, rivaling the MBE growth rates commonly employed in industry for the growth of semiconductor thin films. This order of magnitude increase is achieved by co-supplying Sr from a conventional effusion cell and titanium-tetra-isopropoxide (TTIP) as titanium and oxygen source[25,33]. At such high-growth rates the self-regulated growth window, i.e., the range of TTIP pressures for a given constant Sr flux at which the film lattice parameter does not expand to accommodate unintentional incorporation of excess cations, remains accessible, which is mapped using in situ reflection high-energy electron-diffraction (RHEED) and confirmed by ex situ X-ray diffraction (XRD) measurements, atomic force microscopy (AFM), and scanning transmission electron microscopy (STEM). This record high-growth rate demonstrated experimentally is merely limited by size and stability of the Sr effusion cell used in the research MBE reactor. The trend extrapolated from the data projects that growth rates of about 9 µm h$^{-1}$ are possible if industrial scale MBE hardware is employed instead.

## Results

**Mapping the stoichiometric growth window at high-growth rate.** The conditions to access the self-regulated growth of stoichiometric SrTiO$_3$ were determined for a range of growth rates by observing specific surface reconstructions in real time using in situ RHEED[34,35]. This method avoids multiple film growths and time consuming lattice parameter measurements using ex situ XRD[36,37], and makes accessing the self-regulated growth window a production-line compatible process. This accelerated mapping of growth conditions is key to ensure proper growth condition in a single calibration run in a timely manner, as shown in Fig. 2. In this method, a SrTiO$_3$ film was grown on a (La$_{0.18}$Sr$_{0.82}$)(Al$_{0.59}$Ta$_{0.41}$)O$_3$ (LSAT) single-crystal substrate. The Sr flux supplied from the effusion cell was held constant at $2.50 \times 10^{13}$ cm$^{-2}$ s$^{-1}$ during the entire calibration run, measured using a quartz crystal monitor at the sample position prior to growth. The TTIP flux, given by the beam equivalent pressure (BEP) $p_{TTIP}$ measured at the sample position using an ion gauge, was altered throughout the growth by changing the gas inlet pressure. The color code in Fig. 2a indicates whether stoichiometric (green), Sr-rich (blue), or Ti-rich (purple) growth condition were present. Stoichiometric growth conditions were initially chosen and Sr and TTIP were co-supplied for 10 min, resulting in a 12 nm-thick template layer. The TTIP flux was then increased and growth was continued for 5 min. This procedure was repeated and RHEED images were taken in real time. Figure 2a, b shows the growth sequence along with atomic-resolution STEM micrographs of the calibration sample in cross-section, respectively. STEM imaging revealed that for layers grown under Ti-rich growth conditions small Ti excess was incorporated as point defects, which gave rise to a gradual increase in background intensity relative to the stoichiometric regions of the calibration sample. For growth conditions with larger Ti excess the formation of a Ti-rich second phase was observed. In the case of Sr-rich growth conditions

STEM images revealed the local formation of SrO Ruddlesden–Popper stacking faults propagating throughout the film. SrTiO$_3$ layers were grown within the self-regulated window subsequently on both types of nonstoichiometric layers to determine the minimum thickness required to outgrow the local nonstoichiometry. While stoichiometric SrTiO$_3$ could be repeatedly achieved on layers grown under Ti-rich conditions irrespective of its thickness and degree of Ti-excess, the thickness needed to recover a stoichiometric growth front on layers grown under Sr-rich conditions was about 15 nm. The STEM images further showed that defects from Ti-rich and Sr-rich regions propagated up to about 5 nm into the subsequently grown layer, shown by regions of higher intensity protruding from the fifth layer into the subsequently grown stoichiometric layer shown in Fig. 2b. This marks the minimum growth duration needed to ensure that growth conditions were accurately assessed by surface sensitive RHEED and subsequent layers were not influenced by potential nonstoichiometries of the preceding layer. The RHEED pattern taken in situ during iterative deposition cycles are shown in Fig. 2c, and in more detail in the Supplementary Fig. 3. In the Sr-rich regime, the RHEED pattern was diffuse with a 2-fold reconstruction along the 110 azimuth, which can be easily distinguished from the stoichiometric growth conditions, characterized by a sharp RHEED pattern and loss of the 2-fold reconstruction along 110 as well as the appearance of a 2-fold reconstruction along the 100. For Ti-rich conditions RHEED patterns were diffuse and exhibited a gradual loss of the 2-fold reconstruction along the 110[35,36,38]. Therefore, the growth conditions correlated with the surface reconstruction seen in RHEED and can be used to map the self-regulated growth window on a single calibration sample. The entire growth calibration procedure contained ten different TTIP fluxes and a recovery growth ("reset") layer after mapping the edge of the growth window towards the Ti-rich growth conditions. The growth of the

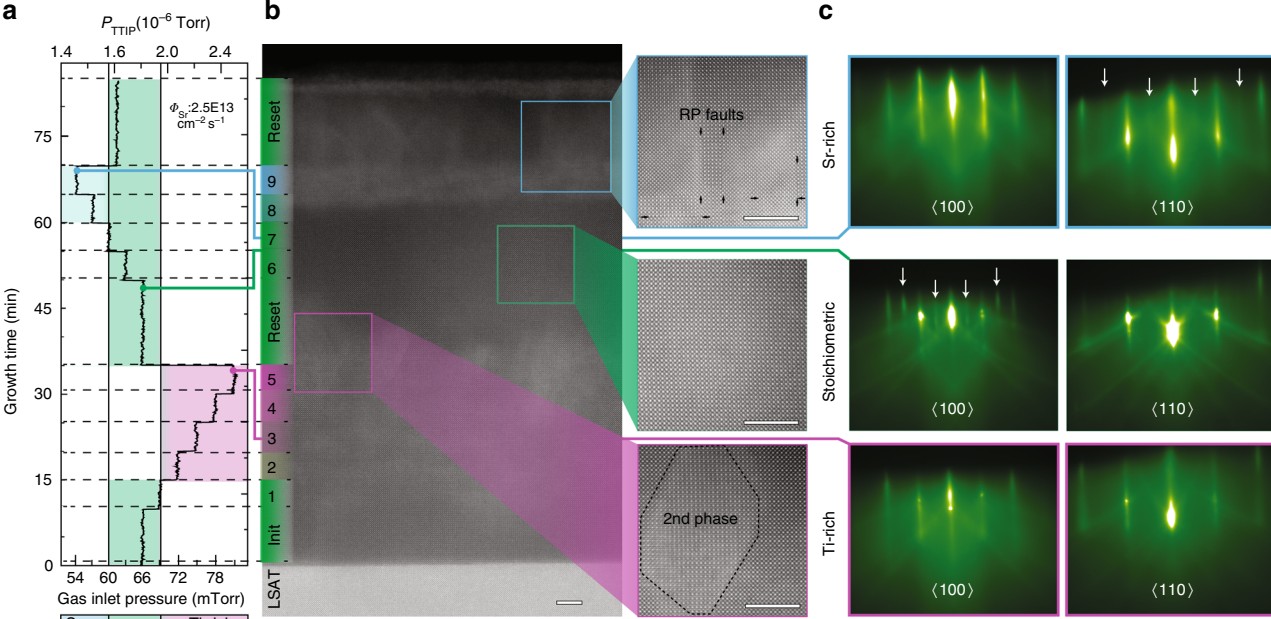

**Fig. 2** Mapping growth conditions using production-line compatible process. Calibration sample using a fixed Sr flux of $2.50 \times 10^{13}$ cm$^{-2}$ s$^{-1}$ grown on (La$_{0.18}$Sr$_{0.82}$)(Al$_{0.59}$Ta$_{0.41}$)O$_3$ (LSAT). **a** Titanium tetraisopropoxide (TTIP) flux sequence using the beam equivalent pressure $p_{TTIP}$ as flux measure. **b** High-angle annular dark-field scanning transmission electron microscopy (HAADF-STEM) cross-section images of the calibration sample. A higher background intensity was found for layers grown under nonstoichiometric condition. The magnified images illustrate the type of defect formed and how far they protruded into the subsequent layer grown under stoichiometric conditions. The precise position of the Ruddlesden Popper (RP) faults indicated by the arrows were deduced from strain maps shown in Supplementary Fig. 9. **c** Reflection high-energy electron diffraction (RHEED) images taken in real-time during growth along the ⟨100⟩ and ⟨110⟩ azimuth reflecting the stoichiometry of the growth front. Scale bars on the HAADF images are 5 nm

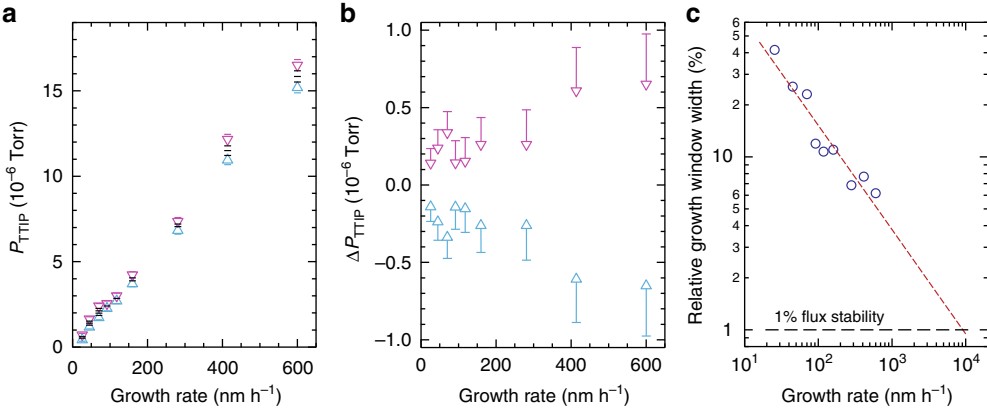

**Fig. 3** Scaling growth window. **a** Titanium tetraisopropoxide (TTIP) beam equivalent pressures $p_{TTIP}$ as a function of film growth rate for which a growth window was accessed. The Sr flux was kept fixed while the TTIP flux was modulated. The black bars represent $p_{TTIP}$ values within the growth window, while the upper red and lower blue triangle indicate the edges of the growth window determined by reflection high-energy electron diffraction (RHEED). The red and blue bar above and below the respective triangles are the first $p_{TTIP}$ values for Ti-rich and Sr-rich growth conditions, respectively. **b** Absolute growth window width $\Delta p_{TTIP}$ as a function of growth rate. The triangles mark the edges of the growth window, while the bars are the values at which a nonstoichiometric growth front was observed in RHEED. **c** Relative growth window width $\Delta p_{TTIP}/p_{TTIP}$ as a function of growth rate, where $p_{TTIP}$ is the TTIP beam equivalent pressures at the growth window center. The line is extrapolated based on the existing data set, projecting that a growth rate of $(9.1 \pm 1.6)$ µm h$^{-1}$ can be achieved at a given Sr flux stability of 1%

calibration sample took about 70 min, which is considerably shortened with increasing growth rate.

**Accelerated growth rate of SrTiO₃.** Rapid growth condition mapping as shown in Fig. 2 was applied to find the self-regulated growth window for Sr-fluxes ranging from $1.25 \times 10^{13}$ cm$^{-2}$ s$^{-1}$ up to $2.50 \times 10^{14}$ cm$^{-2}$ s$^{-1}$. The width and position of the growth window as a function of growth rate is shown in Fig. 3. For a Sr flux of $1.25 \times 10^{13}$ cm$^{-2}$ s$^{-1}$ (growth rate 25.6 nm h$^{-1}$), the TTIP BEP at the growth position had to be kept between 4.28 and $7.10 \times 10^{-7}$ Torr to maintain self-regulated growth, which shifted to $1.52–1.65 \times 10^{-5}$ Torr as the Sr flux was increased to $2.50 \times 10^{14}$ cm$^{-2}$ s$^{-1}$ (growth rate 600.5 nm h$^{-1}$) supplied from two Sr effusion cells operated in tandem. The actual position of the growth window changed linearly with growth rate, which was determined by the Sr flux supplied. The calibration curve of TTIP BEP $p_{TTIP}$ and gas inlet pressure is shown in Supplementary Fig. 4, while the change in growth rate with supplied Sr flux is shown in Supplementary Fig. 5. The absolute width of the growth window $\Delta p_{TTIP}$ increased at higher growth rates (Fig. 3b), whereas the relative width $\Delta p_{TTIP}/p_{TTIP}$ (Fig. 3c) decreased. The latter trend can be extrapolated to determine the growth rate at which unintentional drifts from the Sr effusion cell can still be compensated via the self-regulated adsorption mechanism. Assuming a Sr flux drift of about 1% the growth window would be sufficiently wide up to a growth rate of $(9.1 \pm 1.6)$ µm h$^{-1}$, as shown in Fig. 3c. This growth rate does not mark the highest possible growth rate that can be achieved using hybrid MBE, since the growth window width has been found to increase with growth temperature[26].

To confirm stoichiometric growth conditions from film lattice parameter and surface morphology measurements using ex situ XRD and AFM, nominally 45-nm-thick SrTiO₃ films were grown on LSAT(100) substrates. Films grown within the growth window showed a step-like terrace structure, a root mean square surface roughness of less than 1 nm and absence of crystalline islands irrespective of growth rate (Supplementary Fig. 6). Figure 4a shows high-resolution $2\theta–\omega$ X-ray scans around the 002 SrTiO₃ film and LSAT substrate peaks. The out-of-plane lattice parameter was found to be $3.930 \pm 0.003$ Å irrespective of growth rate, which matches values previously reported for stoichiometric

SrTiO₃ films grown on LSAT[39]. While critical film thickness of SrTiO₃ on LSAT substrates was only 180 nm[40], a ~500-nm-thick SrTiO₃ film was grown homoepitaxially within the growth window for a growth rate of 600 nm h$^{-1}$. $2\theta–\omega$ X-ray scans of the 002 SrTiO₃ film and substrate peak shown in Fig. 4b revealed that they are indistinguishable, as expected for a homoepitaxially grown stoichiometric film[41]. RHEED intensity oscillations are shown in Fig. 4c, d for SrTiO₃ films grown at 40 and 600 nm h$^{-1}$ on LSAT, respectively. In both film growths RHEED intensity oscillations were visible which dampened out with time, indicating an initial layer-by-layer growth which transitioned into a step-flow growth mode. The growth of 10 monolayers of SrTiO₃ took only 23.8 s, in excellent agreement with the growth rate determined from film thickness measurements. Details of the X-ray fits to determine physical film thickness and intrinsic film lattice parameter are detailed in Supplementary Fig. 7.

**Integration on silicon for virtual substrates**. To directly prove the compatibility of high SrTiO₃ film growth rates with the integration on Si wafers 125-nm-thick SrTiO₃ thin films were grown and the results are shown in Fig. 5. Two different Sr sources were used in tandem; a Sr effusion cell with low flux $(2.50 \times 10^{13}$ cm$^{-2}$ s$^{-1})$ to nucleate and initialize the growth on Si using a buffer layer[37], while a second Sr cell calibrated to a much higher flux $(1.00 \times 10^{14}$ cm$^{-2}$ s$^{-1})$ for the film growth at a high rate of 240 nm h$^{-1}$. The film thickness was chosen to avoid film cracking due to the thermal mismatch between Si and SrTiO₃. Wide range XRD scans confirmed single crystalline (001) oriented SrTiO₃ films with an out-of-plane lattice parameter of 3.889 Å, consistent with stoichiometric SrTiO₃ films grown on Si at 850 °C, albeit at much lower growth rates of only 50 nm h$^{-1}$[35,37]. Figure 5b, c shows the rocking curve of the 002 SrTiO₃ film peak with a full width at half maximum of 0.28° along with the reciprocal space map of SrTiO₃ 103 in excellent agreement with those reports[35,37]. Figure 5d shows the film surface morphology of SrTiO₃ on Si grown at 240 nm h$^{-1}$, which showed a stepped terrace morphology with a root mean square surface roughness value less than 1 nm, making it an ideal starting surface for subsequent epitaxial growth, thus fulfilling yet another requirement of a metamorphic buffer layer. Excellent film thickness uniformity across the entire 3-in wafer was evident

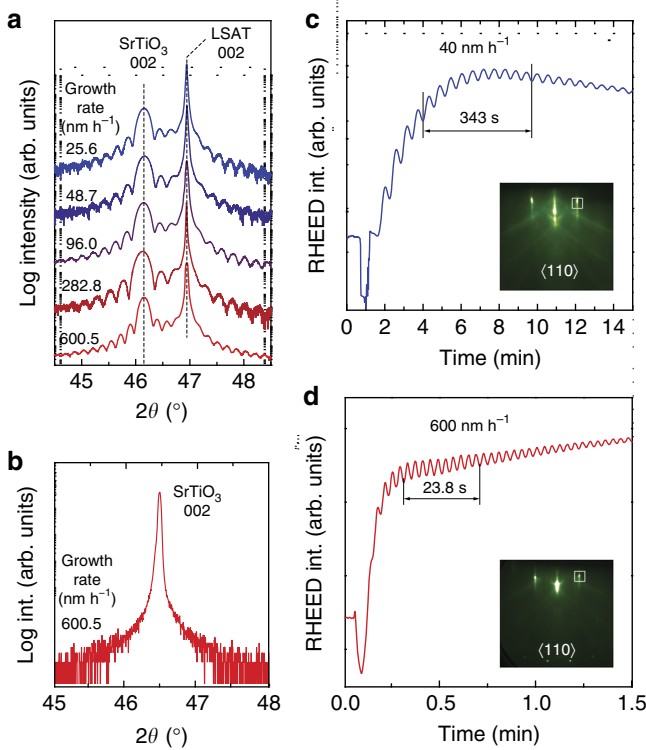

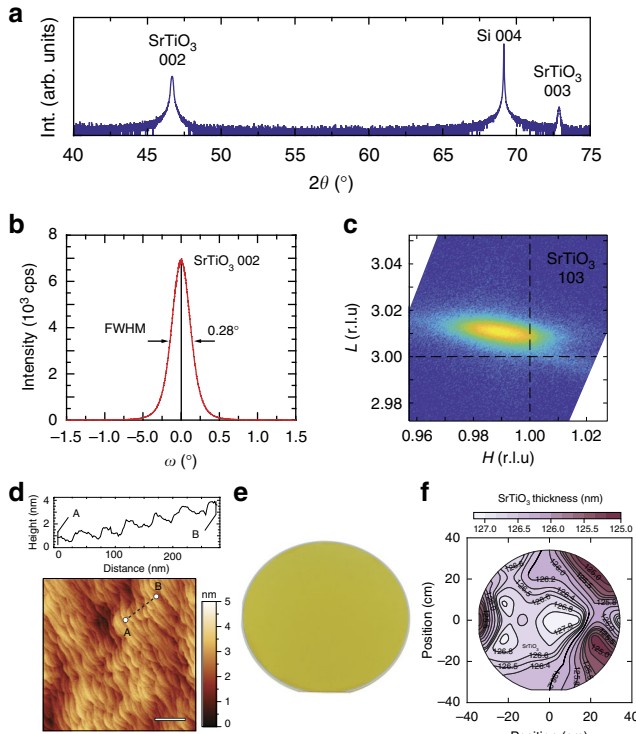

**Fig. 4** X-ray diffraction and RHEED oscillation. **a** High resolution $2\theta$-$\omega$ X-ray scans of the 002 $(La_{0.18}Sr_{0.82})(Al_{0.59}Ta_{0.41})O_3$ (LSAT) substrate (~46.9°) and $SrTiO_3$ film peaks (~46.2°) for growth rates ranging from 25.6 to 600.5 nm h$^{-1}$. **b** $2\theta$-$\omega$ X-ray scan of the 002 peak for a 500-nm-thick $SrTiO_3$ film grown homoepitaxially at 600 nm h$^{-1}$. **c, d** Reflection high-energy electron diffraction (RHEED) intensity oscillation of the 10 diffraction streak (see inset) for $SrTiO_3$ films grown on LSAT at 40 and 600 nm h$^{-1}$. The sharp RHEED intensity drop is caused by the sample shutter operation needed to start the growth, blocking the electron beam

**Fig. 5** Virtual perovskite oxide substrate on Si. **a** Wide range $2\theta$-$\omega$ X-ray scan of a ~126-nm-thick $SrTiO_3$ film grown on 3 in Si wafer at a growth rate of 240 nm h$^{-1}$. **b** Rocking curve ($\omega$-scan) of the 002 $SrTiO_3$ peak with a full width at half maximum (FWHM) of 0.28°. **c** Reciprocal space map of the 103 $SrTiO_3$ film peak in reciprocal lattice units of stoichiometric $SrTiO_3$. **d** Atomic force microscopy image of the $SrTiO_3$ on Si with "epi-ready" step terrace morphology and a surface roughness (RMS) of ~1 nm. A line profile of the film is shown above. Scale bar on AFM image is 200 nm. **e** Photograph of the perovskite oxide virtual substrate, a $SrTiO_3$ metamorphic buffer layer on a 3-in Si wafer. The $SrTiO_3$ film is transparent, the yellow coloration is from interference due to its finite thickness ($\lambda/4$). **f** Wafer scale map of $SrTiO_3$ film thickness on Si obtained from spectroscopic ellipsometry with a smaller than 1% film thickness variation across the wafer

from optical inspection (Fig. 5e) and the $SrTiO_3$ film thickness variation was determined across the entire wafer using spectroscopic ellipsometry, see Fig. 5f. A less than 1% change in $SrTiO_3$ thickness was extracted using an optical model detailed in Supplementary Fig. 8. Cross-sectional STEM and energy dispersive spectroscopy in Supplementary Fig. 10 show a single-crystalline $SrTiO_3$ film on the silicon substrate, separated by a 2–4 nm amorphous $SiO_x$ interfacial layer. Secondary ion mass spectrometry (SIMS) of $SrTiO_3$ on silicon is given in Supplementary Fig. 11, revealing carbon incorporation from the metalorganic precursor is limited primarily to the interface due to the decreased cracking efficiency of the metalorganic precursor at low temperatures. Carbon contamination is minimized for the bulk of the deposition at higher temperatures, in agreement with previous studies[42].

## Discussion

These growth experiments demonstrate that it is possible to integrate functional oxides in a scalable and economic way on Si, addressing a long-standing challenge to synthesize high-quality perovskite materials on an industrial scale. The accessibility of the self-regulated growth window at high-growth rates afforded by hybrid MBE and the compatibility of this growth process with Si enables functional diversification of electronic devices in the current More than Moore era. The wide-ranging properties of perovskite oxides will open up routes to augment additional

functions to existing devices. The scalability of growth rates enables the development of a metamorphic buffer technology for virtual perovskite oxide substrates. Rather than being limited to single crystal substrates with sizes of up to only two inches using bulk crystal growth techniques such as Czochralski or Bridgman–Stockbarger methods, perovskite oxides on Si wafers with excellent crystalline quality and epi-ready surface morphologies provide more cost effective substrates and thus will enable more growth experiments. The availability of virtual substrates will drive innovation at the fundamental level and speed up the process development from basic scientific discoveries to system level maturity, thus boosting research productivity and technological significance of functional epitaxial oxide thin films. While accessing a self-regulated growth window at high growth rates was demonstrated for $SrTiO_3$ as a prototypical perovskite oxide, many $ABO_3$ perovskites ($ABO_3$, A = Ca[43], Sr[40], Ba[44], Gd[45], Nd[46], Sm[47] with B = Ti; A = Sr[48], Ca[49], La[50] with B = V; as well as $LaAlO_3$[25], $BaSnO_3$[51], and $SrRuO_3$[52]) have already been grown by hybrid MBE. This suggests that scaling growth rates is a general feature of this growth approach rather than being limited to $SrTiO_3$, indicating that the development of virtual substrates for perovskite oxides can be expanded to other perovskite oxides in an economic, scalable way as well.

## Methods

**Thin film growth.** SrTiO₃ films were grown on LSAT, SrTiO₃, and Si substrates using a DCA M600 hMBE deposition system equipped with 2 Sr dual filament solid source thermal effusion cells with 60 cc crucibles. A gas delivery system was used to supply Ti in form of the volatile metalorganic molecule TTIP without a carrier gas. Sr fluxes were calibrated prior to growth using a quartz crystal monitor located at the sample growth position. The TTIP gas inlet pressure was monitored using a capacitance manometer in the injector system coupled to a linear leak valve to adjust and maintain the flux of TTIP to the substrate. Molecular oxygen was supplied and maintained at an oxygen background pressure of $\sim 3.0 \times 10^{-7}$ Torr. Growth windows were mapped out for fixed Sr fluxes between $1.25–25.0 \times 10^{13}$ cm$^{-2}$ s$^{-1}$, which were achieved by operating the Sr cells between 410 and 560 °C. A single Sr effusion cell was found to become unstable at fluxes exceeding $2.00 \times 10^{14}$ cm$^{-2}$ s$^{-1}$. Therefore, for high-growth rate experiments, a second Sr effusion cell was used in tandem to achieve a sufficiently large and stable flux. The reported growth rate was determined from film thickness measurements determined by fitting Laue oscillations of XRD data using GenX[53]. All SrTiO₃ films on LSAT were grown at a temperature of 900 °C measured by the substrate heater thermocouple. SrTiO₃ thin films were grown on Si wafers at 850 °C using the process detailed in ref. [35], followed by a post growth anneal, described in ref. [37]. Silicon wafers were etched in 10% buffered oxide etch, rinsed in deionised water, and placed under vacuum within 10 min of the initial etch. A one-half monolayer of Sr was deposited at 600 °C, followed by annealing at 720 °C for 10 min to remove any additional oxygen before deposition. The sample was cooled to 400 °C, and 4 monolayers of SrTiO₃ were deposited by co-supplying Sr using the standard cell calibrated to flux of $2.5 \times 10^{13}$ cm$^{-2}$ s$^{-1}$ and TTIP at a baratron pressure of 52 mTorr at a growth rate of $\sim$50 nm h$^{-1}$. The temperature was then raised to 600 °C and an additional 8 monolayers of SrTiO₃ were deposited at the same fluxes to complete the buffer layer. The temperature was further increased to 850 °C and a 10 nm layer of SrTiO₃ was grown using the standard Sr flux and TTIP baratron pressure of 67 mTorr to account for the increased desorption rate at elevated temperatures. Elemental oxygen was then introduced to the chamber during this stage to a background pressure of $3 \times 10^{-7}$ Torr. The second effusion cell calibrated to $1.0 \times 10^{14}$ cm$^{-2}$ s$^{-1}$ was used to deposit the remainder of the film a growth rate of 240 nm h$^{-1}$, with a TTIP baratron pressure of 137 mTorr. After growth, samples were annealed in a furnace at 850 °C in air to improve crystalline quality and relieve built-in strain in the films.

**Atomic force microscopy.** The surface morphology of SrTiO₃ films grown at different growth rates on LSAT and Si were measured using a Bruker Dimension Icon atomic force microscope (AFM) operated in peak force tapping mode.

**X-ray diffraction.** High resolution of XRD of films grown within the growth window was carried out using a Phillips X'Pert Panalytical Pro operating with Cu-K$_{\alpha 1}$ radiation. $2\theta$–$\omega$ scans were taken with an acceptance angle of 180″, while rocking curve was recorded using a divergence angle of 26″ an acceptance angle of 35″. Reciprocal space map was taken on the same system with a divergence angle of 26″, and recorded using a Panalytical Pixel3D detector.

**Transmission electron microscopy.** Cross-sectional high-resolution STEM was carried out using a FEI Titan operated at 200 kV accelerating voltage. TEM sample preparation of films on LSAT substrates consisted of mechanical polishing at wedge angle of 2° to a thickness of <5 μm, followed by thinning to electron transparency using a Gatan PIPSII Ar-ion milling along [100]. Samples on silicon were prepared using the Scios DualBeam focused ion beam using Ga⁺ ions. Accelerating voltage was scaled from 30 keV for the initial lamella preparation, and scaled down to 2 keV for the final sample thinning.

**Spectroscopic ellipsometry.** Ellipsometric data, i.e., phase difference Δ and amplitude ratio Ψ in the spectral range from 0.75–3.0 eV were collected from 25 different locations across the wafer (see details in Supplementary Fig. 8) using a J. A. Woollam Co., Inc. M-2000XI single rotating compensator spectroscopic ellipsometer in reflection mode at 3 different angles of incidence, 50°, 60°, and 70°. The thickness values and refractive index for the SrTiO₃ film were extracted by means of a least squares regression analysis employing the Levenberg–Marquardt algorithm. An optical model consisting of a Si substrate (semi-infinite)/native silicon oxide interface layer/SrTiO₃ film/SrTiO₃ film surface roughness was used to extract the physical parameters. The surface roughness was represented by a Bruggeman effective medium approximation of 0.5 void + 0.5 film material fractions. For the substrate and silicon native oxide layer, the silicon[54] and silicon dioxide[54] material parameters were used. The parameterization of the refractive index of the SrTiO₃ film included a combination of the high-frequency dielectric constant and two Sellmeier oscillators. Oscillator parameters as well as thicknesses of the SrTiO₃ film, the native oxide film, and the surface roughness layer were used as fitting parameters.

**Secondary ion mass spectrometry.** SIMS of SrTiO₃ on silicon was performed using a PHI nano TOF. A primary 30 keV Bi₃⁺ ion beam was rastered over an area of $100 \times 100$ μm. The etch rate was approximately 3 nm cycle$^{-1}$.

## Data availability

The data that support the findings of this study are available from the corresponding author upon reasonable request.

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

## Acknowledgements

J.M.L. and R.E.H. acknowledge National Science Foundation through the Penn State MRSEC program DMR-1420620, J.R. acknowledges DMR-1629477 and support through the NSF graduate student fellowship, M.B. and R.E.H. acknowledge the Department of Energy (Grant DE-SC0012375), L.Z. acknowledges the National Science Foundation through DMR-1352502. We thank Dr. Arnab Sen Gupta for assisting in growth of samples, Profs. Jon-Paul Maria and Venkat Gopalan, as well as Drs. Craig Eaton and Julian Walker for helpful discussions.

## Author contributions

J.L., M.B., L.Z. and R.E.H. conceived the project. J.L., J.R. and M.B. performed the growth and characterization of SrTiO₃ films on LSAT, and SrTiO₃ at high-growth rates, as well as XRD and AFM characterization of all samples. J.L. and L.Z. grew SrTiO₃ films on silicon. A.P. measured the film thickness using spectroscopic ellipsometry, J.L. prepared and imaged the HR-STEM samples. J.L. and R.E.H. co-wrote the paper with input and suggestions from all authors.

## Additional information

**Competing interests:** The authors declare no competing interests.

