## [Peer Review File · Nature Communications]

Reviewers' comments:

Reviewer #1 (Remarks to the Author):

The manuscript describes the authors' scaling up of the hybrid MBE technique for growing SrTiO₃ that they developed to rates exceeding 600 nm/hr. The growth technique itself is no longer new but the demonstration of scalability to growth rates amenable to production makes this work of great interest to the thin film growth community. The authors did a thorough job in demonstrating that their process works over a wide range of growth rates and that the film quality is very good. I recommend that this manuscript be published in Nature Communications if the following corrections/clarifications are made:

1) The authors claim their process is adsorption-controlled growth. However, their process is not truly adsorption-controlled or self-regulating as excess Ti can still form in the film if the TTIP pressure is too high. They have a more relaxed growth window compared to solid source MBE but they should not say that this process is adsorption-controlled or self-regulated.

2) The authors should describe in more detail how the nucleation of STO on Si is done using their process. Their process works well for growing on oxide surfaces but something special is probably done for the nucleation on Si to work.

3) Can the authors comment on the degree of carbon incorporation in the STO films they grew? It is not immediately obvious that TTIP decomposes fully to TiO₂ on contact with the substrate. TTIP being an ALD precursor typically requires water to break up the ligands. How much carbon gets incorporated in the film?

4) Can the authors state how much SiO₂ is formed for their process? At 850C, STO is quite permeable to oxygen and SiO₂ is likely slowly but continuously forming during the entire growth process. The post-growth anneal probably adds even more SiO₂. A cross-section TEM image would be nice for thick STO grown on Si at high rate.

5) It should be clarified that the initial study as shown in Fig. 1 was done using an LSAT substrate. It is currently not stated in the manuscript although one can deduce it from the TEM image.

Reviewer #2 (Remarks to the Author):

The authors mainly reported the fabrication of SrTiO₃ films at rapid growth rate (exceeding 600 nm/hour) by hybrid molecular oxide epitaxy. Such rapid growth rates were claimed applicable to grow uniform and stoichiometric SrTiO₃ films on Si wafer.

My main concern is that the novelty of the manuscript may not satisfy the standard of Nature communication. It is true that the increase of the growth rate up to tenfold is a technical breakthrough. However, the manuscript doesn't give deep explanation on the underlying mechanism, nor discover new phenomena during the fabrication. As a result, it doesn't seem to me I would recommend it to a more technical journal.

Meanwhile, I have a few questions/comments to the authors:

1. In Fig. 2(b), the RP faults were labeled manually by black lines. The authors may consider to add some structural illustration next to the STEM image. On the current image, it is not obvious which is the right phase, which is the RP faults.
2. Since the major point of the manuscript is about the rapid growth of SrTiO₃ on Si, the authors should definitely show the STEM image of SrTiO₃ on Si. This will help the readers to get more straightforward view on the structure of the film.
3. In Fig. 5(d), the authors claimed the AFM image "showed a stepped terrace morphology", which is not clear to me where is the step and terrace. The authors should replace this image with a more distinct one.
4. When the authors grew SrTiO₃ on Si wafer, they first grew a 6 nm thick buffer followed by a 10 nm crystalline SrTiO₃ layer, with slow growth rate ~ 50 nm/hr. Then the growth rate was subsequently increased to 240 nm/hr to grow the rest of the film (the increase is only about 5 times). The question is: what if the SrTiO₃ is grown directly with the rapid growth rate? If the initial growth with slow rate is indeed inevitable, then isn't it more fair to calculate the 'average growth rate', instead of only counting the fast growth period?

Reviewer's Comments and Authors' Response

We would like to start by thanking all reviewers for their valuable comments concerning our manuscript and the many insightful suggestions, all of which have been addressed in the revised manuscript to improve upon the original submission. A detailed reply to all reviewer's comments is given below in blue. Changes in the manuscript were in orange.

Reviewer #1:

The manuscript describes the authors' scaling up of the hybrid MBE technique for growing SrTiO₃ that they developed to rates exceeding 600 nm/hr. The growth technique itself is no longer new but the demonstration of scalability to growth rates amenable to production makes this work of great interest to the thin film growth community. The authors did a thorough job in demonstrating that their process works over a wide range of growth rates and that the film quality is very good. I recommend that this manuscript be published in Nature Communications if the following corrections/clarifications are made:

1) The authors claim their process is adsorption-controlled growth. However, their process is not truly adsorption-controlled or self-regulating as excess Ti can still form in the film if the TTIP pressure is too high. They have a more relaxed growth window compared to solid source MBE but they should not say that this process is adsorption-controlled or self-regulated.

Answer:

We partially agree with this comment. The described growth process is certainly not a truly adsorption-controlled growth mechanism in the sense of the classical example of GaAs grown by conventional molecular beam epitaxy (MBE), in which even large quantities of excess As can desorb at sufficiently high substrate temperatures due to the large volatility of As. However, even in the case of MBE growth of GaAs conditions can be chosen such that As rich films are grown. Since MBE is a thin film synthesis technique that is carried out under conditions far from thermodynamic equilibrium nonstoichiometric films can be created, e.g. by sufficiently low substrate temperatures and high As overpressures so that the incorporation is governed by the kinetics at play and not the equilibrium consideration.

In the specific case employed here - the growth of SrTiO₃ by hybrid MBE - the self-regulated process is limited to a specific range of TTIP pressures, as pointed out correctly by the reviewer. Indeed, the key difference is that while TTIP undergoes a thermal decomposition process while being adsorbed on the growth surface, resulting in the formation of a non-volatile reactant which cannot desorb from the surface, this is not the case for As. Nevertheless, a growth window for SrTiO₃ by hybrid MBE is present. The fact that the growth window can be accessed using hybrid MBE is key to achieve such high growth rates and maintain the cation stoichiometry. Accessing the solid source MBE window for SrTiO₃ has been demonstrated to be impractical, see e.g. Appl. Phys. Lett. 95, 032906 (2009); <https://doi.org/10.1063/1.3184767>. In conventional solid source MBE this would be only possible if growth conditions were chosen such that SrO becomes the volatile constituent of the film, which would require very high substrate temperatures and very low growth rates. We therefore disagree with the statement that hybrid MBE provides a '... a more relaxed growth window compared to solid source MBE...'

Under these considerations we interpret the reviewer's comment that he/she is concerned readers might interpret the statements of adsorption-controlled or self-regulated mechanism in the sense of the classical example GaAs. We therefore have changed the main manuscript and stress that the self-regulation growth mechanism was only present in the growth window. We further clearly define the meaning of the growth window as range of TTIP pressures for a given constant Sr flux and substrate temperature at which the film lattice parameter does not expand to accommodate unintentional incorporation of excess cations early in the manuscript (page 3). The sentence was changed to:

At such high growth rates the self-regulated growth window, *i.e. the range of TTIP pressures for a given constant Sr flux at which the film lattice parameter does not expand to accommodate unintentional incorporation of excess cations*, remained accessible, which was mapped using *in-situ* reflection high-energy electron-diffraction (RHEED) and confirmed by *ex-situ* X-ray diffraction (XRD) measurements, atomic force microscopy (AFM), and scanning transmission electron microscopy (STEM).

2) The authors should describe in more detail how the nucleation of STO on Si is done using their process. Their process works well for growing on oxide surfaces but something special is probably done for the nucleation on Si to work.

Answer:

The nucleation of SrTiO₃ on Si requires a special procedure that is different from simply growing SrTiO₃ on a perovskite substrate. The first procedure to successfully grow SrTiO₃ on Si by hybrid MBE has been published by our group in 2014, (published here <https://doi.org/10.1002/pssr.201409383>) and has been refined over the years. We have employed the recipe first published in early 2018, see <https://pubs.acs.org/doi/abs/10.1021/acsnano.7b07539>. We have added all details for the nucleation of SrTiO₃ on Si (highlighted in orange) to the methods section for thin film growth, which now reads:

Thin Film Growth: SrTiO₃ films were grown on LSAT, SrTiO₃, and Si substrates using a DCA M600 *h*MBE deposition system equipped with 2 Sr dual filament solid source thermal effusion cells with 60cc crucibles. A gas delivery system was used to supply Ti in form of the volatile metalorganic molecule titanium tetraisopropoxide (TTIP) without a carrier gas. Sr fluxes were calibrated prior to growth using a quartz crystal monitor (QCM) located at the sample growth position. The TTIP gas inlet pressure was monitored using a capacitance manometer in the injector system coupled to a linear leak valve to adjust and maintain the flux of TTIP to the substrate. Molecular oxygen was supplied and maintained at an oxygen background pressure of $\sim 3.0 \times 10^{-7}$ Torr. Growth windows were mapped out for fixed Sr fluxes between $1.25\text{--}25.0 \times 10^{13}$ cm⁻² s⁻¹, which were achieved by operating the Sr cells between 410 and 560°C. A single Sr effusion cell was found to become unstable at fluxes exceeding 2.00×10^{14} cm⁻² s⁻¹. Therefore, for high growth rate experiments, a second Sr effusion cell was used in tandem to achieve a sufficiently large and stable flux. The reported growth rate was determined from film thickness measurements determined by fitting Laue oscillations of X-ray diffraction data using GenX⁵³. All SrTiO₃ films on LSAT were grown at a temperature of 900 °C measured by the substrate heater thermocouple. SrTiO₃ thin films were grown on Si wafers at 850°C using the process

detailed in Ref. ³⁵, followed by a post growth anneal, described in Ref. ³⁷. Silicon wafers were etched in 10% buffered oxide etch, rinsed in DI water, and placed under vacuum within 10 minutes of the initial etch. A ½ monolayer of Sr was deposited at 600°C, followed by annealing at 720°C for 10 minutes to remove any additional oxygen before deposition. The sample was cooled to 400°C, and 4 monolayers of SrTiO₃ were deposited by co-supplying Sr using the standard cell calibrated to flux of 2.5×10^{13} atoms/cm²s and TTIP at a baratron pressure of 52 mTorr at a growth rate of ~50 nm/hr. The temperature was then raised to 600°C and an additional 8 monolayers of SrTiO₃ were deposited at the same fluxes to complete the buffer layer. The temperature was further increased to 850°C and a 10 nm layer of SrTiO₃ was grown using the standard Sr flux and TTIP baratron pressure of 67 mTorr to account for the increased desorption rate at elevated temperatures. Elemental oxygen was then introduced to the chamber during this stage to a background pressure of 3×10^{-7} Torr. The second effusion cell calibrated to 1.0×10^{14} atoms/cm²s was used to deposit the remainder of the film a growth rate of 240 nm/hr, with a TTIP baratron pressure of 137mtorr. After growth, samples were annealed in a furnace at 850°C in air to improve crystalline quality and relieve built-in strain in the films.

3) Can the authors comment on the degree of carbon incorporation in the STO films they grew? It is not immediately obvious that TTIP decomposes fully to TiO₂ on contact with the substrate. TTIP being an ALD precursor typically requires water to break up the ligands. How much carbon gets incorporated in the film?

Answer:

Carbon incorporation into SrTiO₃ films grown by hybrid MBE has been a big concern from the very beginning. A very thorough analysis has been published by Jalan and Stemmer in 2009 (J. Vac. Sci. Technol. A 27, 1365 (2009); <https://doi.org/10.1116/1.3253355>) which has shown that carbon incorporation is very low if growth is performed at high temperature. Specifically, for growth temperatures of 800°C or higher SrTiO₃ films were found to only contain a few ppm of carbon ($\sim 5 \times 10^{17}$ cm⁻³), while at lower temperature (725°C) carbon impurity concentration was $\sim 2 \times 10^{18}$ cm⁻³. Except for the nucleation layer of SrTiO₃, which was deposited at 400°C, the SrTiO₃ films were grown on Si at a substrate temperature of 850°C. Therefore, we expect only spurious traces of carbon in the film. We have performed secondary ion mass spectrometry (SIMS) on 120-nm-thick SrTiO₃ films grown on Si and added the results in the supplement (Suppl. Fig. 11). In good agreement with the previous results by Jalan et.al. a small carbon signal slightly above the detection limit was found. A larger carbon concentration was found at the film/substrate interface which we attributed to the low growth temperature of the nucleation layer. In the SIMS profile the carbon peak at the beginning of the etch cycles is not from adventitious carbon typically found on the film surface from air exposure when transferring the sample from the MBE growth reactor to the SIMS analysis chamber. This increase of carbon concentration arose from a sputter stop for about 20 min in which the freshly sputtered SrTiO₃ surface was exposed to the SIMS chamber background pressure of 3E-8 Torr. Exposing the freshly sputtered SrTiO₃ film surface gave rise to a spurious contamination, yet a measurable signal which was over an order of magnitude higher than the detection limit of the SIMS experiment. This further indicated the low carbon concentration in the grown SrTiO₃ film. We have added the SIMS profile as Fig 11 to the Supplement (Suppl. Fig. 11):

4) Can the authors state how much SiO₂ is formed for their process? At 850C, STO is quite permeable to oxygen and SiO₂ is likely slowly but continuously forming during the entire growth process. The post-growth anneal probably adds even more SiO₂. A cross-section TEM image would be nice for thick STO grown on Si at high rate.

Answer:

A scanning transmission electron microscopy image along with high resolution high angle annular dark field (HAADF) images of the interface region as well as atomic resolution energy dispersive X-ray spectroscopy (EDXS) maps for Sr, Ti, O, Si and C were taken. The interface layer was found to be about 4 nm, the brighter contrast closer to the SrTiO₃ film indicated a higher concentration of a heavy element, which was identified by the EDXS maps as Sr. Note the sharp transition that was found for titanium and oxygen and the two contrast changes for Si. We have added a cross-sectional STEM images and EDXS maps as Fig 10 to the supplement (Suppl. Fig. 10):

5) It should be clarified that the initial study as shown in Fig. 1 was done using an LSAT substrate. It is currently not stated in the manuscript although one can deduce it from the TEM image.

Answer:

We have clarified this by updating the Figure caption of Figure 1. It now reads:

Figure 1| Growth rates and control of film stoichiometry. Intrinsic SrTiO₃ film lattice parameter reported for different growth rates using scalable oxide thin film growth techniques: pulsed laser deposition (PLD), sputtering, metal-organic chemical vapor deposition (MOCVD), atomic layer deposition (ALD), molecular beam epitaxy (MBE) and hybrid molecular beam epitaxy (hMBE). In all cases except hybrid MBE the degree of film nonstoichiometry increased with growth rate. The number close to data points shown refer to references detailed in Suppl. Fig. 1. The defect concentration δ of Sr-rich (Sr_{1- δ} TiO₃) and Ti-rich (Sr_{1+ δ} TiO₃) films due to nonstoichiometric growth condition was determined from the intrinsic film lattice parameter expansion using a calibration curve given in Suppl. Fig. 2. **A growth rate of 0.6 $\mu\text{m/hr}$ for stoichiometric SrTiO₃ by hybrid MBE was achieved for the growth on LSAT substrates.**

Reviewer #2:

The authors mainly reported the fabrication of SrTiO₃ films at rapid growth rate (exceeding 600 nm/hour) by hybrid molecular oxide epitaxy. Such rapid growth rates were claimed applicable to grow uniform and stoichiometric SrTiO₃ films on Si wafer.

My main concern is that the novelty of the manuscript may not satisfy the standard of Nature communication. It is true that the increase of the growth rate up to tenfold is a technical breakthrough. However, the manuscript doesn't give deep explanation on the underlying mechanism, nor discover new phenomena during the fabrication. As a result, it doesn't seem to I would recommend it to a more technical journal.

Meanwhile, I have a few questions/comments to the authors:

1) In Fig. 2(b), the RP faults were labeled manually by black lines. The authors may consider to add some structural illustration next to the STEM image. On the current image, it is not obvious which is the right phase, which is the RP faults.

Answer:

We have revised Fig. 2(b) and added a Figure in the Supplement (Suppl. Fig. 9) to further clarify the position of the RP faults. The change in Fig. 2b is that rather by overlaying a line over the RP fault we indicate the two ends by arrows. We agree that RP stacking faults cannot be easily seen directly in HAADF-STEM, but are readily observed in linear strain maps since they cause an increase in the local strain. We therefore added a Suppl. Figs 9 to illustrate the RP in further detail, including a structural illustration. The revised Fig. 2(b) of the main manuscript and the new Fig. in the supplement (Suppl. Fig. 10) are shown below

Fig. 2 main manuscript

Suppl. Fig. 10

2. Since the major point of the manuscript is about the rapid growth of SrTiO₃ on Si, the authors should definitely show the STEM image of SrTiO₃ on Si. This will help the readers to get more straightforward view on the structure of the film.

Answer:

Thanks for the suggestion, we have added a cross-sectional STEM along with high resolution high angle annular dark field (HAADF) images of the interface region as well as atomic resolution energy dispersive X-ray spectroscopy (EDXS) maps for Sr, Ti, O, Si and C in the supplement of the manuscript (Suppl. Fig. 10). Dotted lines in the Figure indicate the interfaces of the amorphous SiO₂ layer with crystalline Si and SrTiO₃, respectively. The interface layer was found to be about 4 nm, the brighter contrast closer to the SrTiO₃ film indicated a higher concentration of a heavy element, which was identified by the EDXS maps as Sr. Note the sharp transition that was found for titanium and oxygen and the two contrast changes for Si.

3. In Fig. 5(d), the authors claimed the AFM image “showed a stepped terrace morphology”, which is not clear to me where is the step and terrace. The authors should replace this image with a more distinct one.

Answer:

We have replaced the previous AFM image by another AFM scan taken on the same sample and hope that the image is much clearer now. We have also added a line profile. Note the step-height between terraces, i.e. regions at a constant height, of about 1 nm, suggesting step bunching of the films, which is attributed to the relatively large miscut of the Si wafer. The revised Fig. 5 of the main manuscript is shown below:

4. When the authors grew SrTiO₃ on Si wafer, they first grew a 6 nm thick buffer followed by a 10 nm crystalline SrTiO₃ layer, with slow growth rate ~ 50 nm/hr. Then the grow rate was subsequently increased to 240 nm/hr to grow the rest of the film (the increase is only about 5 times). The question is: what if the SrTiO₃ is grown directly with the rapid growth rate? If the

initial growth with slow rate is indeed inevitable, then isn't it more fair to calculate the 'average growth rate', instead of only counting the fast growth period ?

Answer:

For the growth of SrTiO₃ on Si we decided to report the highest growth rate achieved throughout the growth. Note that the quoted 240 nm/hr is much lower than the 600 nm/hr we demonstrated for the growth of SrTiO₃ on LSAT. The highest growth rate of 600 nm/hr experimentally demonstrated was also limited by the maximum stable flux generated by the two Sr cells operated in tandem. Unfortunately, the size and design of the Sr effusion cells used in the experiments were not optimized for a large Sr flux supply and did not allow for stable Sr fluxes for values higher than $2 \times 10^{14} \text{ cm}^{-2} \text{ s}^{-1}$.

The reason for the much lower growth rate for SrTiO₃ on Si was that we could not provide the fluxes from both Sr cells for the rest of the film grown at high growth rate since we had to use one of the Sr effusion cells at a much lower flux to grow the nucleation and buffer layer to ensure good epitaxy of SrTiO₃ on Si. While we think that the buffer layer can also be successfully grown at a larger growth rate than demonstrated here, it would have required to identify new growth parameter for the nucleation and buffer layer at 400°C and 600°C. It is indeed an important question whether a slower growth rate for the initial steps to epitaxially integrate SrTiO₃ on Si is inevitable or not, but exploring this interesting question is outside the scope of the work presented here.

We would also like to add another note: if the flux stability of the Sr effusion cells would not have been the limiting factor in the experiment, the SrTiO₃ film grown after the nucleation step could have been grown at 600 nm/hr instead of 240 nm/hr. In that case the total growth time for the 120-nm-thick SrTiO₃ film on Si would have been 12.5 minutes for the nucleation and buffer layer (12 monolayers, i.e. about 5nm, and additional 10 nm were grown at a growth rate of 50 nm/hr) and 10.5 minutes for the remaining 105 nm; an average growth rate of a little over 300 nm/hr, which would be still be higher than the quoted 240 nm/hr.

REVIEWERS' COMMENTS:

Reviewer #1 (Remarks to the Author):

The additional clarifying statements in the main text and two new supplemental figures address all my concerns from my original review. The new version is suitable for publication in Nature Communications.

Reviewer #2 (Remarks to the Author):

The authors have reasonably replied to my comments. I have no more questions and recommend it for publication.